# Time-Considerable Dialogue Models via Reranking by Time Dependency

**Yuiko Tsunomori[1], Masakazu Ishihata[1], Hiroaki Sugiyama[1]**
[1]NTT Communication Science Laboratories
{yuiko.tsunomori,masakazu.ishihata,hiroaki.sugiyama}@ntt.com

## Abstract

In the last few years, generative dialogue models have shown excellent performance and have been used for various applications. As chatbots become more prevalent in our daily lives, more and more people expect them to behave more like humans, but existing dialogue models do not consider the time information that people are constantly aware of. In this paper, we aim to construct a *time-considerable* dialogue model that actively utilizes time information. First, we categorize responses by their naturalness at different times and introduce a new metric to classify responses into our categories. Then, we propose a new reranking method to make the existing dialogue model *time-considerable* using the proposed metric and subjectively evaluate the performances of the obtained time-considerable dialogue models by humans.[1]

## 1 Introduction

In the last few years, generative dialogue models have achieved outstanding performance ([Ziegler et al., 2019](#); [Adiwardana et al., 2020](#); [Ouyang et al., 2022](#); [Thoppilan et al., 2022](#)) and have been used in various applications, including search engines, recommendations, healthcare, finance, and more ([Ling et al., 2023](#)). As chatbots permeate our daily lives, more and more people expect chatbots to behave in a human-like manner. Examples of research to make chatbots more human-like include the introduction of common sense ([Wang et al., 2020](#)), empathy ([Ma et al., 2020](#)), personas ([Zhang et al., 2018](#)), and so forth. The common point among these studies is that they have achieved richer dialogue by actively utilizing not only internal information obtained through the conversation but also *external information* that does not appear in the current dialogue. On the other hand, *time information*,

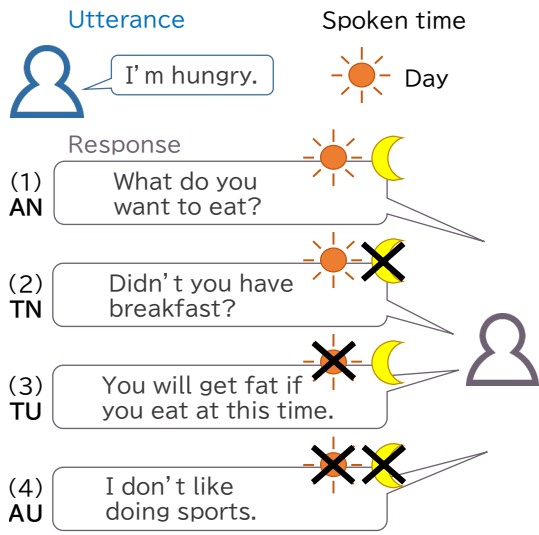

Figure 1: Conversation examples depending on time.

the most basic and important external information, still does not seem to be considered important in dialogue models.

Humans are basically always aware of time in conversation, whether explicitly or implicitly, because the *naturalness* of utterances and responses may change based on their spoken times (e.g., time of the day, day of the week, and season). Given an utterance and its spoken time, a response can be categorized into the following four types by focusing on the time variations of its naturalness:

- **AN**: always natural
- **TN**: temporarily natural at the spoken time
- **TU**: temporarily unnatural at the spoken time
- **AU**: always unnatural

Figure [1](#) shows examples of the above four types of responses. Assume two time periods, day and night, where responses (1)-(4) have different levels of naturalness at different times. Responses (1) and (2) ($N \triangleq AN \cup TN$) are natural to a given utterance at a spoken time, although not (3) and (4)

---

[1]The detailed information about our dataset is available at https://github.com/nttcslab/time-considerable-dialogue-model

($\mathbf{U} \triangleq \mathbf{TU} \cup \mathbf{AU}$). On the other hand, the naturalness of (2) and (3) ($\mathbf{T} \triangleq \mathbf{TN} \cup \mathbf{TU}$) changes with the spoken time (day or night), and that of (1) and (4) ($\mathbf{A} \triangleq \mathbf{AN} \cup \mathbf{AU}$) remains unchanged. Hereafter in this paper, we refer to such categorization of responses as the *NUTA categories*.

For a dialogue model to achieve natural conversation, it is expected to generate natural responses $\mathbf{N}$ and to avoid generating unnatural responses $\mathbf{U}$. If a dialogue model correctly evaluates the naturalness of responses considering the spoken time, we call it *time-aware*; otherwise, we call it *time-unaware*. Many existing dialogue models are time-unaware because they are trained on datasets *without* time information. As a result, a time-unaware dialogue model may consider $\mathbf{TU}$ responses as natural as $\mathbf{N}$ responses because they are natural at some time. In other words, it may generate a $\mathbf{TU}$ response, which is inappropriate at the spoken time. The simplest way to construct a time-aware dialogue model is to train a general dialogue model with time information (Sato et al., 2017). Recently, the output of a large language model (LLM) was adjusted by giving an appropriate prompt (Lester et al., 2021; Bae et al., 2022; Liu et al., 2022), and prompting the time information to an LLM is another promising way to achieve a time-aware dialogue model. The difference between time-unaware and time-aware dialogue models is that the former considers a response as natural if it is natural at some time, while the latter does if it is natural at its spoken time; namely, time-aware dialogue models avoid generating $\mathbf{TU}$ responses.

Prior research has shown that users' impressions of dialogue models are improved by *actively* utilizing external information (Vinyals and Le, 2015; Li et al., 2016; Zhou et al., 2021). Referring to this fact, we assume that users' impressions will be similarly improved by actively utilizing time information; namely, *users prefer $\mathbf{TN}$ to $\mathbf{AN}$* (we empirically verify this assumption in Section 3). Under this assumption, we aim to realize a *time-considerable* dialogue model that actively outputs more $\mathbf{TN}$ than $\mathbf{AN}$. The difference between time-aware and time-considerable models is that the former only considers time information to evaluate the naturalness of responses at a given spoken time (*time-aware naturalness*), while the latter actively generates responses whose naturalness varies with time. To realize time-considerable models, we need a new criterion to distinguish between $\mathbf{TN}$

and $\mathbf{AN}$ and a new mechanism to generate $\mathbf{TN}$ responses.

In this paper, we propose a new reranking method that is a post-processing method to make existing dialogue models more time-considerable. As a preliminary analysis, we first verify our assumption *users tend to prefer $\mathbf{TN}$ to $\mathbf{AN}$* by human evaluation in Section 3. We next formally define the NUTA categories and propose an automatic metric for the NUTA categories in Section 4. In Section 5, we propose a new reranking method using the proposed automatic metric to make existing dialogue models time-considerable and subjectively evaluate obtained time-considerable models to verify whether our reranking method improves response qualities.

## 2 Related Work

Previous studies have pointed out that general dialogue models trained on large-scale datasets tend to generate neutral (bland, generic, and hackneyed) responses (Li et al., 2016; Serban et al., 2016). To tackle this issue, some dialogue models utilizing *external information* to generate interesting and informative responses have been proposed, where examples of external information include system/user persona (Zhang et al., 2018; Roller et al., 2021; Lu et al., 2022), knowledge graphs (Zhang et al., 2020), knowledge sources (Parthasarathi and Pineau, 2018; Majumder et al., 2022), interpersonal relationships (Utami and Bickmore, 2019), and situated environments (Misu, 2018). Some studies have empirically shown that generating responses specific to external information improved users' impression of the dialogue models (Vinyals and Le, 2015; Li et al., 2016; Zhou et al., 2021). In this paper, we introduce the time information as external information and consider that $\mathbf{T}$ responses, whose naturalness varies with time, are specific to time information. Furthermore, to verify whether the time information improves model performance as well as other external information, we evaluate the qualities of $\mathbf{T}$ responses by human evaluations.

In this paper, we define a dialogue model as *time-aware* if the model evaluates the naturalness of responses considering their spoken times. Sato et al. (2017) proposed a time-aware dialogue model that is an encoder-decoder model based on Long-Short Term Memory (LSTM) (Zaremba et al., 2015) inspired by Johnson et al. (2017) and trained on utterance-response pairs with their timestamps ex-

tracted from Twitter. Another recent technique to achieve a time-aware dialogue model is *prompting* LLMs, where prompting is a technique to guide LLMs in generating high-quality and relevant responses by providing detailed descriptions and/or input-output examples of the target task as input (Brown et al., 2020). Various prompting-based methods for utilizing external information have been proposed, and their examples include system/user persona information (Kasahara et al., 2022; Lee et al., 2022), knowledge sources (Liu et al., 2022), and fictional character's style (Han et al., 2022). In this paper, we aim to construct a *time-considerable* dialogue model, which actively utilizes the time information, and propose a new reranking method to make the existing dialogue models more time-considerable using time-aware naturalness represented by a time-aware dialogue model. While time-aware dialogue models only consider time to evaluate the naturalness of responses, time-considerable dialogue models actively generate responses whose naturalness varies with time.

## 3 Preliminary Analysis

Through this paper, we assume *users prefer* **TN** *to* **AN**, and in this section, we verify this assumption by human evaluations. We first constructed a NUTA dataset consisting of the tuples of utterances, their spoken times, their responses, and their NUTA categories: **AN**, **TN**, **TU**, and **AU**. Then, we conducted a subjective evaluation to determine which category of responses users found most interesting and informative.

### 3.1 The NUTA Dataset

We constructed a NUTA dataset, which is a collection of tuples $\mathbf{t} \triangleq \langle u, t, r, c \rangle$, where $u$ is an utterance, $t \in \mathcal{T} \triangleq \{0, 1, \ldots, 23\}$ is its spoken time in 24-hour time format, $r$ is a response to $u$, and $c \in \{\mathbf{AN}, \mathbf{TN}, \mathbf{TU}, \mathbf{AU}\}$ is its NUTA category.

We first prepared a set of utterances $\mathcal{U}$. We extracted Japanese tweets posted between May and December 2022 with filtering rules described in Appendix A.1 and randomly selected 1,000 tweets. We manually deleted tweets containing discriminatory, violent, or other inappropriate expressions. As a result, we obtained 640 appropriate tweets and used them as utterances $\mathcal{U}$.

We next obtained responses to the prepared utterances $\mathcal{U}$ by crowdsourcing, where we used

Lancers[2], a Japanese crowdsourcing service. We assigned one crowd worker to each utterance $u \in \mathcal{U}$ and asked them to perform the following tasks to create responses:

1. Create response $r_u^{\mathbf{AN}}$ to $u$ that is natural at any time.
2. Select two time periods $t^{\mathbf{N}}$ and $t^{\mathbf{U}}$ ($t^{\mathbf{N}}, t^{\mathbf{U}} \in \mathcal{T}$) and create response $r_u^{\mathbf{TN}}$ to $u$ that is natural at $t^{\mathbf{N}}$ but not at $t^{\mathbf{U}}$.

For instance, given utterance $u$ = "It seems the train is stopped," the crowd worker selected two time periods $t = 22$ and $t' = 6$ and created two responses $r_u^{\mathbf{AN}}$ = "Really? I wonder when it will start moving." and $r_u^{\mathbf{TN}}$ = "Wow, it's almost the last train. I wonder what's going to happen?"

Finally, we constructed a NUTA dataset by creating the following four tuples for each utterance $u \in \mathcal{U}$, where $u'$ was randomly chosen from $\mathcal{U}$ so that $u \neq u'$:

$$\langle u, t^{\mathbf{N}}, r_u^{\mathbf{AN}}, \mathbf{AN} \rangle, \quad \langle u, t^{\mathbf{N}}, r_u^{\mathbf{TN}}, \mathbf{TN} \rangle,$$
$$\langle u, t^{\mathbf{U}}, r_u^{\mathbf{TN}}, \mathbf{TU} \rangle, \quad \langle u, t^{\mathbf{N}}, r_{u'}^{\mathbf{AN}}, \mathbf{AU} \rangle.$$

Thus, the dataset consists of 2,560 ($= |\mathcal{U}| \times 4$) tuples. Table 1 shows examples of four created tuples for the same utterance.

### 3.2 Subjective Evaluation

We conducted a subjective evaluation to determine which response category is the most interesting and informative for humans.

We introduced a new metric to measure the quality of responses considering time information. The metric is in the range $[0, 1]$ and based on the Sensibleness, Specificity, Interestingness (SSI) metric (Thoppilan et al., 2022) for evaluating responses based on context. Our metric, denoted by SSI-t, averages the following four scores:

- *Sensibleness for the utterance* (SU): If its spoken time is ignored, is the response reasonable to its utterance?
- *Sensibleness for the spoken time* (ST): If its utterance is ignored, is the response reasonable to its spoken time?
- *Specificity to time* (S): Is the response specific to any time regardless of its spoken time?
- *Interestingness* (I): Is the response interesting or informative?

---

[2]https://www.lancers.jp/

| Utterance $u$ = It seems the train is stopped. (電車止まってるらしい‥) | | |
|---|---|---|
| Time $t$ | Response $r$ | Category $c$ |
| 22 | Really? I wonder when it will start moving. (そうなんだ。いつになったら動くのかなー。) | **AN** |
| 22 | Wow, it's almost the last train. I wonder what's going to happen? (ええー。あと少しで終電だけど、どうなるのかな？) | **TN** |
| 6 | Wow, it's almost the last train. I wonder what's going to happen? (ええー。あと少しで終電だけど、どうなるのかな？) | **TU** |
| 22 | Understood! I'll decide on the character as soon as possible. (了解！なるべく早くキャラを決めるわ) | **AU** |

Table 1: Four tuples $\mathbf{t} = \langle u, t, r, c \rangle$ for the same utterance. A crowd worker selected $t^{\mathbf{N}} = 22$ and $t^{\mathbf{U}} = 6$ and generated two sentences $r_u^{\mathbf{AN}}$ and $r_u^{\mathbf{TN}}$. The utterance and all responses were originally written in Japanese and translated into English by the authors.

For example, a score SU of 1.0 indicates $r$ is a perfect response to $u$ if $t$ is ignored, and a score S of 0.0 indicates the naturalness of $r$ never changes over time.

We randomly selected 25 tuples for each NUTA category (100 tuples in total). For any tuple $\mathbf{t} = \langle u, t, r, c \rangle$ and any score $SC \in \{$SU, ST, S, I$\}$, we asked two expert annotators, who are in-house workers specialized in annotating dialogues and have worked in their positions for at least five years, to rate $SC$ of $\mathbf{t}$ with either 0 or 1. We defined the $SC$ value of $\mathbf{t}$ as the average of two obtained rates and the SSI-t value of $\mathbf{t}$ as the average of all $SC$ values of $\mathbf{t}$. Finally, for any NUTA category $c$ and any score $SC \in \{$SU, ST, S, I, SSI-t$\}$, we obtained the $SC$ value of $c$ by averaging those of all tuples $\mathbf{t}$ whose categories are $c$. Table 2 shows four examples of tuples and their obtained values.

Table 3 shows the SSI-t scores for each NUTA category and indicates that **TN** achieved the highest quality (SSI-t): more specifically, the highest ST and I scores. We believe that the results support our assumption that *users prefer* **TN** *to* **AN**, and based on this assumption, we will propose a method to realize a time-considerable dialogue model that actively outputs more **TN** than **AN**.

## 4 Automatic Metric for NUTA Categories

We propose a new automatic metric for classifying the NUTA categories of given responses and experimentally show that our metric can correctly categorize the responses of the NUTA dataset.

### 4.1 Definition

We mathematically introduce the *time-aware naturalness* and the *time dependency* of responses and define the NUTA categories using those quantities. Let $u$, $r$, and $t$ be an utterance, a response to $u$, and the time at which the conversation took place.

**Time-aware naturalness** We assume that the *time-aware naturalness* (TAN) of $u$ and $r$ at $t$ is *implicitly* defined by conditional probability distribution $p(u, r \mid t) = p(u \mid t)\, p(r \mid u, t)$, where $p(u \mid t)$ and $p(r \mid u, t)$ indicate the TANs of $u$ at $t$ and $r$ given $u$ at $t$. We consider $\mathbf{N}$ responses of the NUTA categories as natural at spoken time $t$; i.e., response $r$ is classified as $\mathbf{N}$ (resp. $\mathbf{U}$) iff $p(r \mid u, t)$ is high (resp. low). Since it is very difficult to know the true TAN $p$, throughout this paper, we assume that TAN $p$ is given as a time-aware dialogue model that allows us to evaluate $p(r \mid u, t)$ for any $u$, $r$, and $t$.

**Change of naturalness** Using TAN $p$, we define the change of (log) naturalness (CN) from $t'$ to $t$ as

$$\mathrm{CN}_{t':t}(u, r) \triangleq \ln \frac{p(u, r \mid t)}{p(u, r \mid t')}, \quad (1)$$

$$\mathrm{CN}_{t':t}(u) \triangleq \ln \frac{p(u \mid t)}{p(u \mid t')}, \quad (2)$$

$$\mathrm{CN}_{t':t}(r \mid u) \triangleq \ln \frac{p(r \mid u, t)}{p(r \mid u, t')}. \quad (3)$$

By definition, the following equation must hold:

$$\mathrm{CN}_{t':t}(u, r) = \mathrm{CN}_{t':t}(u) + \mathrm{CN}_{t':t}(r \mid u), \quad (4)$$

where the CN of conversation $(u, r)$ can be factorized into the CNs of $u$ and $r$ given $u$. For instance, $\mathrm{CN}_{t':t}(u, r) > 0$ holds iff the conversation $(u, r)$ is more natural at $t$ than at $t'$.

**Time dependency** Using the above CN, we define *time dependency* (TD) of $r$ given $u$ as

$$\mathrm{TD}(r \mid u, t) \triangleq \max_{t^{\mathbf{U}} \in \mathcal{T}} \mathrm{CN}_{t^{\mathbf{U}}:t}(r \mid u), \quad (5)$$

$$\mathrm{TD}(r \mid u) \triangleq \max_{t^{\mathbf{N}} \in \mathcal{T}} \mathrm{TD}\big(r \mid u, t^{\mathbf{N}}\big). \quad (6)$$

$\mathrm{TD}(r \mid u, t)$, denoted by **TD@t**, is the CN of $r$ given $u$ from the most unnatural time $t^{\mathbf{U}}$ to spoken

| Utterance $u$ | Time $t$ | Response $r$ | Cat. $c$ | SU | ST | S | I | SSI-t |
|---|---|---|---|---|---|---|---|---|
| If gummies are within reach, I can't stop eating them endlessly. (グミ、手の届くとこにあると無限に食べ続けてしまう。) | 10 | I also love gummies, especially the ones with lots of fruit juice. (グミ私も好きです。特に果汁多めのやつが) | **AN** | 1.0 | 1.0 | 0.0 | 1.0 | 0.75 |
| Does anyone want to play Apex Legends ranked match with me sometime? (エペランクマ今度一緒にやってくれる人いませんか?…) | 17 | Maybe we can play a game together tonight. (せっかくだから今晩やってもいいですよ) | **TN** | 1.0 | 1.0 | 1.0 | 1.0 | 1.00 |
| Oh! I just realized I have over 1000 followers! (え!!今気づいたんだけどふぉろわさん1000人いってる!?) | 10 | Congrats! I'm glad to hear the good news before I go to bed. (おめでとう!寝る前に良いニュース聞けて嬉しいよ。) | **TU** | 1.0 | 0.0 | 1.0 | 0.5 | 0.63 |
| Maybe I'll eat curry today. (今日はカレー食うかな) | 17 | Autumn goes by so fast. (秋はあっという間にすぎていくよね) | **AU** | 0.0 | 1.0 | 0.0 | 0.5 | 0.38 |

Table 2: Four example tuples $\langle u, t, r, c \rangle$ and their obtained SU, ST, S, I, and SSI-t values. All utterances and responses were originally written in Japanese and translated into English by the authors.

| Category | SU | ST | S | I | SSI-t |
|---|---|---|---|---|---|
| **AN** | 0.98 | 0.86 | 0.14 | 0.66 | 0.66 |
| **TN** | 0.97 | 1.00 | 0.80 | 0.76 | 0.89 |
| **TU** | 0.93 | 0.19 | 0.88 | 0.71 | 0.68 |
| **AU** | 0.06 | 0.87 | 0.13 | 0.66 | 0.43 |

Table 3: SU, ST, S, I, and SSI-t scores of responses of each NUTA category

| Category | $p(r \mid u, t)$ | $TD(r \mid u)$ |
|---|---|---|
| **AN** | High | Low |
| **TN** | High | High |
| **TU** | Low | High |
| **AU** | Low | Low |

Table 4: Definition of NUTA categories using TAN $p(r \mid u, t)$ and TD@all $TD(r \mid u)$.

time $t$, which evaluates whether $r$ is specific to spoken time $t$. Consequently, $r$ has a high TD@$t$ if it is natural at $t$ but unnatural at another time and a low TD (i.e., near zero) if its naturalness remains unchanged as time changes. On the other hand, $TD(r \mid u)$, denoted by TD@all, is the CN from the most unnatural time $t^{\mathbf{U}}$ to the most natural time $t^{\mathbf{N}}$, which evaluates whether $r$ is specific to time or not. So, $r$ has a high TD@all if it is natural at some time but not at another time. Since we consider the naturalness of **T** responses of the NUTA categories varies with time, $r$ is classified as **T** (resp. **A**) iff $TD(r \mid u)$ is high (resp. low).

**NUTA category** By the definitions of TAN $p(r \mid u, t)$ and TD@all $TD(r \mid u)$, we define each NUTA category by Table 4. For instance, response $r$ is classified as **TN** iff both $p(r \mid u, t)$ and $TD(r \mid u)$ are high, and $r$ is classified as **AU** iff both $p(r \mid u, t)$ and $TD(r \mid u)$ are low. Strictly speaking, to use this definition, two thresholds must be set that distinguish between the high and low of $p(r \mid u, t)$ and $TD(r \mid u)$; however, since we believe that determining these thresholds in advance is difficult, all the methods proposed in this paper are designed so that they do *not* require such thresholds.

**Related criteria** In natural language processing, various criteria have been proposed for measuring the dependency between two sentences. Li et al. (2016) proposed *pointwise mutual information* (PMI) to choose appropriate response $r$ to given utterance $u$:

$$\text{PMI}(r \mid u) \triangleq \log \frac{p(r \mid u)}{p(r)}. \quad (7)$$

As an extension of PMI, Paranjape and Manning (2021) proposed *pointwise conditional mutual information* (PCMI) to cope with additional external information other than utterance $u$ to evaluate an appropriate response to $u$. Given utterance $u$, response $r$, and time information $t$ as external information, PCMI@$t$ is defined as

$$\text{PCMI}(r \mid u, t) \triangleq \log \frac{p(r \mid u, t)}{p(r \mid u)}, \quad (8)$$

where let $p(r \mid u)$ be the *time-unaware naturalness* (TUN). While TD@$t$ in Eq. (5) is the CN of $r$ given $u$ from the most unnatural time $t^{\mathbf{U}}$ to the current time $t$, PCMI@$t$ is the CN of $r$ given $u$ when the naturalness changes from TUN to TAN at $t$. In a similar manner as TD@all, we define PCMI@all as $\text{PCMI}(r \mid u) \triangleq \max_{t^{\mathbf{N}} \in \mathcal{T}} \text{PCMI}(r \mid u, t^{\mathbf{N}})$.

| | Naturalness | | TD | | PCMI | |
|---|---|---|---|---|---|---|
| | TUN | TAN | @all | @t | @all | @t |
| **N** | 0.60 | **0.68** | 0.47 | 0.58 | 0.46 | 0.55 |
| **T** | 0.54 | 0.54 | **0.68** | 0.60 | 0.53 | 0.48 |

Table 5: Accuracy of **N/U** and **T/A** classifications of each quantity. Best scores are indicated by **bold**.

| | TD | | PCMI | |
|---|---|---|---|---|
| | @all | @t | @all | @t |
| **AN** | **0.54** | 0.43 | 0.43 | 0.38 |
| **TN** | 0.48 | **0.60** | 0.35 | 0.51 |
| **TU** | **0.44** | 0.19 | 0.34 | 0.14 |
| **AU** | **0.45** | 0.44 | 0.33 | 0.32 |
| Ave. | **0.48** | 0.42 | 0.36 | 0.34 |

Table 6: Accuracy of each NUTA category of each combination. Best scores are indicated by **bold**.

We believe our TD is a more appropriate metric for time information than PCMI. PCMI considers the presence or absence of time information, not its change; however, time information always exists and changes, unlike such common external information as user persona and knowledge graphs. Therefore, PCMI is expected to be more blurred in its evaluation than TD. For instance, suppose two time ranges, $t_1$ and $t_2$, such that $p(t_1) = p(t_2) = 0.5$ where $(u, r)$ is a strongly time-specific response such that $p(r \mid u, t_1) = 0$ (i.e., $r$ cannot be a response to $u$ at $t_1$) and $p(r \mid u, t_2) = 1$ (i.e., $r$ is a perfect response to $u$ at $t_2$). Then, $p(r \mid u) = 0.5$ since $p(r \mid u) = \sum_{t \in \{t_1, t_2\}} p(t) p(r \mid u, t)$. By Eqs. (3), (5) and (6), $\mathrm{TD}(r \mid u) = \infty$ where TD@all considers $r$ as a strongly time-specific response. On the other hand, $\mathrm{PCMI}(r \mid u, t) = \ln 2 \approx 0.69$ where PCMI@all considers $r$ as not so time-specific. Thus, using TUN $p(r \mid u)$ blurs the evaluation of time dependency, and our TD@all is expected to detect time-specific responses more clearly than PCMI@all.

## 4.2 Experiments

We conducted experiments to show that our automatic metric for the NUTA categories correctly orders responses of the NUTA dataset of Section 3.1.

### 4.2.1 Experimental Settings

For each utterance $u \in \mathcal{U}$, we ranked four tuples containing $u$ in the NUTA dataset by each quantity: TUN $p(r \mid u)$, TAN $p(r \mid u, t)$, TD@all $\mathrm{TD}(r \mid u)$, TD@t $\mathrm{TD}(r \mid u, t)$, PCMI@all $\mathrm{PCMI}(r \mid u)$, and PCMI@t $\mathrm{PCMI}(r \mid u, t)$. We attached a label high (resp. low) to the top (resp. bottom) of two tuples in each ranking for each quantity and used the obtained labels to classify NUTA categories.

As TUN $p(r \mid u)$, we used the Transformer-based Japanese dialogue model (TJD) (Sugiyama et al., 2023) with 1.6B parameters trained on more than two billion tweet-reply pairs: $(u, r)$. We constructed TAN $p(r \mid u, t)$ by fine-tuning the above model using tweet-reply-time triplets: $(u, r, t)$, where the fine-tuning dataset was obtained similarly as (Sato et al., 2017): we collected Japanese tweets with replies from August 2021 to April 2022 with filtering rules described in Appendix A.1 and obtained 470,255,625 triplets. We denote the fine-tuned time-aware TJD by TJD-t and used TJD $p(r \mid u)$ and TJD-t $p(r \mid u, t)$ to evaluate TD@all/t and PCMI@all/t. Detailed implementational settings are shown in Appendix A.2.

### 4.2.2 Experimental Results

We first checked whether TAN $p(r \mid u, t)$ and TD@all $\mathrm{TD}(r \mid u)$ correctly classified **N** and **T**. Using high/low labels obtained by each quantity, we categorized tuples with high (resp. low) labels as **N** (resp. **U**) and computed the accuracy of the **N** category of each quantity. Similarly, we also computed the accuracy of the **T** category. Table 5 shows the accuracies of each quantity and indicates that TAN and TD@all achieved the highest accuracy of **N** and **T**. Consequently, TAN and TD@all are appropriate quantities for evaluating the naturalness and time dependency of responses.

Next, we checked whether the combination of TAN and TD@all correctly classified each NUTA category. We categorized tuples into one of **AN**, **TN**, **TU**, and **AU** according to Table 4 using the combination of high/low labels obtained by TAN and those obtained by one of TD@all, TD@t, PCMI@all, and PCMI@t. Table 6 shows the accuracy of each NUTA category of each combination and indicates that TD@all achieved the best average accuracy; however, for the **TN** category, TD@t achieved the best. Since TD@t evaluates whether response $r$ is specific to the current time $t$ but not to other times, it is more effective to detect **TN** responses than **TU** responses. Because our original motivation for using these quantities as automatic metrics for the NUTA categories is to detect **TN** responses that users prefer than **AN**, we conclude that TD@t $\mathrm{TD}(r \mid u, t)$ is the most appropriate metric for constructing a time-considerable dialogue model.

| Model | len | distinct-1 | distinct-2 | SU | ST | S | I | SSI-t |
|---|---|---|---|---|---|---|---|---|
| TJD | 17.22 | 0.36 | 0.57 | 0.86 | 0.84 | 0.16 | 0.40 | 0.56 |
| TC-TJD | 12.50 | **0.55** | **0.78** | 0.86 | **0.85** | **0.19** | **0.45** | **0.59** |
| TJD-t | 16.74 | 0.33 | 0.48 | 0.78 | 0.83 | 0.12 | 0.36 | 0.52 |
| TC-TJD-t | 14.44 | **0.49** | **0.75** | **0.86** | **0.86** | **0.20** | **0.48** | **0.60** |
| GPT-3.5 | 40.50 | 0.45 | 0.76 | 0.80 | 0.89 | 0.36 | 0.60 | 0.66 |
| TC-GPT-3.5 | 34.92 | 0.44 | 0.74 | 0.74 | **0.90** | **0.53** | 0.54 | **0.68** |
| GPT-4 | 37.79 | 0.40 | 0.73 | 0.86 | 0.96 | 0.89 | 0.62 | 0.83 |
| TC-GPT-4 | 30.88 | 0.37 | 0.69 | 0.83 | 0.96 | **0.93** | 0.59 | 0.83 |

Table 7: Average length (len), distinct-$N$ ($N = 1, 2$), SU, ST, S, I, and SSI-t scores of each model. TC-$\mathcal{M}$ is a time-considerable $\mathcal{M}$ achieved by our proposed reranking method. White and gray rows correspond to $\mathcal{M}$ and TC-$\mathcal{M}$. If TC-$\mathcal{M}$'s score exceeded $\mathcal{M}$' one, it is indicated by **bold**.

## 5 Time-Considerable Dialogue Models

We propose a new reranking method to make existing dialogue models *time-considerable*. We applied our method to various existing models, including GPT-4, which is a state-of-the-art LLM, and evaluated the time-considerable dialogue models to verify whether they improved response qualities.

### 5.1 Proposed Reranking method

Let $\mathcal{M}$ be any dialogue model that can generate multiple responses to the same utterance. Our proposed reranking method extends $\mathcal{M}$ to be time-considerable. Given base dialogue model $\mathcal{M}$, TAN $p(r \mid u, t)$, positive integer $N$, and probability $\delta$, we obtain time-considerable response $r^*$ to utterance $u$ at time $t$ by the following manner:

1. Generate $N$ candidate responses to $u$ at $t$, denoted by $\mathcal{R} \triangleq \{r_i \mid i \in [N]\}$, from base dialogue model $\mathcal{M}$,

2. Evaluate TAN $p(r_i \mid u, t)$ for all $r_i \in \mathcal{R}$ and delete $r_i$ from $\mathcal{R}$ if $r$ has no sufficient naturalness; i.e., $p(r_i \mid u, t) \le \delta$,

3. Evaluate TD@$t$ $\mathrm{TD}(r_i \mid u, t)$ for all $r_i \in \mathcal{R}$ and find the most time-specific response $r^* \in \max_{r \in \mathcal{R}} \mathrm{TD}(r \mid u, t)$,

4. Return obtained $r^*$ as a time-considerable response to $u$ at $t$.

Since Step 2 removes candidate responses with lower naturalness than threshold $\delta$, the filtering mechanism may improve the naturalness of the final response $r^*$ when the response generation model is weak.

Our proposed reranking method is simple but strong because we can create various time-considerable dialogue models by combining existing base dialogue models and TANs, where base dialogue model $\mathcal{M}$ is required only to generate multiple responses for the same utterance and TAN $p(r \mid u, t)$ only to be evaluable. Namely, our method can be applied to dialogue models whose architectures and parameters are not publicly available but are provided as APIs. Of course, if $\mathcal{M}$ is time-aware and evaluable, it can also be used as TAN $p(r \mid u, t)$.

### 5.2 Experiments

We applied our proposed reranking method to existing dialogue models and gauged their performance by human subjective evaluations.

#### 5.2.1 Experimental Settings

We briefly explain our experimental settings, and the detailed settings are shown in Appendix A.2.

**Base dialogue models** We used the following four dialogue models as base dialogue model $\mathcal{M}$ of our proposed reranking method:

1. **TJD** is a transformer-based Japanese dialogue model with 1.6B parameters trained on over two billion tweet-reply pairs (Sugiyama et al., 2023) described in Section 4.2.

2. **TJD-t** is a time-aware TJD obtained by fine-tuning described in Section 4.2.

3. **GPT-3.5** is an extension LLM of GPT-3 (Brown et al., 2020) with 355B parameters and supports various tasks in many languages. gpt-3.5-turbo is a specialized GPT-3.5 for dialogue tasks and is provided as an API.

4. **GPT-4** is a large-scale multimodal model that extends GPT-3.5 (OpenAI, 2023). gpt-4 is a specialized GPT-4 for dialogue tasks and is provided as an API.

All models can generate multiple responses for the same utterance by top-$p$ sampling (Holtzman et al., 2020), where $p$ is a hyperparameter and set to 0.9 through the experiments. Since GPT-3.5 and GPT-4 do not treat time information as input, they are originally time-unaware; however, in our experiment, we used them with prompts to generate time-aware responses. We created prompts based on a sample prompt for dialogue tasks provided by OpenAI and shown in Appendix A.2.

**Settings on proposed reranking method** For any base dialogue model $\mathcal{M}$, we denote time-considerable $\mathcal{M}$ achieved by our proposed reranking method by TC-$\mathcal{M}$ (e.g., TC-TJD, TC-GPT-4). Throughout the experiments, we used TJD-t as TAN $p(r \mid u, t)$ and set $N = 20$ and $\delta = 0$, where $N$ is the number of candidate responses to be generated and $\delta$ is a threshold for filtering candidate responses by their naturalness. We set $\delta = 0$ because we used sufficiently strong models for response generation and did not aim to improve their naturalness.

**Time intervals** Since TD@$t$ is the maximum of $\mathrm{CN}_{t^{\mathbf{U}}:t}(r \mid u)$ for all possible $t^{\mathbf{U}} \in \mathcal{T}$, $t^{\mathbf{U}}$ could be very close to spoken time $t$, and in such cases, the time dependency of $r$ might be not interpretable for humans since it is too *short-term*. To avoid detecting such short-term time dependency, we divided 24 hours into three intervals, morning (3 to 8), noon (9 to 17), and night (18 to 2), and defined their representatives as 6, 13, and 22, with reference to Yamamoto and Shimada (2019). The reason of adopting the above intervals is that morning/noon/night is defined based on the human life-cycle; thus, this interval is more intuitive for human understanding. We evaluated TD@$t$ by Eq. (5) with $\mathcal{T}$ as representatives except for one of spoken time $t$ (e.g., $t = 10 \Rightarrow \mathcal{T} = \{6, 22\}$).

**Evaluation dataset** As an evaluation dataset, we prepared 100 utterance-time pairs in the same manner as $\mathcal{U}$ in Section 3.1, where we excluded the same tweets as the NUTA dataset. Given utterance $u$ and its spoken time $t$, for each base dialogue model $\mathcal{M} \in \{\text{TJD, TJD-t, GPT-3.5, GPT-4}\}$, we generated the best response of $\mathcal{M}$ and a time-considerable response of TC-$\mathcal{M}$, denoted by $\bar{r}_{\mathcal{M}}$ and $r_{\mathcal{M}}^*$. Thus, the evaluation dataset consisted of 100 tuples of $u$, $t$, and responses $\bar{r}_{\mathcal{M}}$ and $r_{\mathcal{M}}^*$ for each $\mathcal{M}$.

**Evaluation criteria** For each dialogue model, we computed the average output response length (len) and distinct-$N$ ($N = 1, 2$), which is the ratio of the number of unique $N$-grams to the total words generated by the dialogue model, for evaluating its diversity (Li et al., 2016). We also obtained the ST, SU, S, I, and SSI-t scores defined in Section 3.2 by human subjective evaluations. We assigned five crowd workers for each response and asked them to evaluate its SU, ST, S, and I scores with a five-point Likert scale (1 to 5) and normalized them into $[0, 1]$ scale. We obtained the SU, ST, S, and I values of each model by averaging the annotated values of responses generated by the model. Finally, we obtained its SSI-t value as the average of those values.

### 5.2.2 Experimental Results

Table 7 shows the values of each model for the eight indices: the average output response length (len), distinct-$N$ ($N = 1, 2$), SU, ST, S, I, and SSI-t scores. For any score except len, a higher value denotes a better result. In the table, if the score of TC-$\mathcal{M}$ exceeds the original $\mathcal{M}$, we highlighted it in **bold**; in other words, bold scores were improved by our proposed reranking method.

TC-$\mathcal{M}$ improved the ST, S, and SSI-t scores for almost all $\mathcal{M}$; namely, it was more time-considerable than the original $\mathcal{M}$. TC-$\mathcal{M}$ tended to prefer a short response to $\mathcal{M}$ because the len of TC-$\mathcal{M}$ is smaller than $\mathcal{M}$ for each model. For GPTs, the distinct-$N$, SU, and I scores of TC-$\mathcal{M}$ were slightly lower than $\mathcal{M}$. This is because those scores of $\mathcal{M}$ were already sufficiently high, and our reranking method aims to improve time-considerability but not diversity and general sensibility of naturalness. Consequently, we confirmed that our reranking method improved the quality of the existing dialogue models.

One interesting observation from this experiment is that TJD-t, which we used as TAN $p(r \mid u, t)$, had lower quality than TJD. This fact suggests that fine-tuning explained in 4.2 was not enough and that our reranking method improved the performance of the existing dialogue models even if TAN's performance is somewhat low. However, of course, using TAN with low performance is not desirable. For GPTs, TC-$\mathcal{M}$ achieved a higher S score but lower SU and I scores than the original $\mathcal{M}$. We believe this deterioration of SU and I scores can be easily avoided using a more accurate model as TAN, such as GPT-4.

| | |
|---|---|
| Utterance | Tomo-san, you can get up feeling refreshed after drinking; I really like you. (tomoさんのお酒飲んだ次の日寝起きいいのすき) @ 10 a.m (daytime) |
| TJD | Tomo-san! Tomo-san! Tomo-san! Tomo-san! (とーもーさん!!とーもーさん!!とーもーさん!!とーもーさん!!とーもーさん!!) |
| TC-TJD | I **overslept and missed my first class**. (**1**限寝過ごしたよ) |
| TJD-t | I also get up feeling refreshed after drinking. (わたしもお酒飲んだ次の日寝起きいいの) |
| TC-TJD-t | I'm sure that **I'll get back to bed**. (二度寝安定ですねこれは) |
| GPT-3.5 | Yes, I have trouble getting up after drinking, but I slept well last night, so I feel fine. (そうですね、お酒を飲んだ翌日は寝起きが悪くなりますが、今日はしっかりと寝ていたので大丈夫です。) |
| TC-GPT-3.5 | Yes, you feel better. But I was drinking last night, so I am a little hungover today. (そうですね、体が軽く感じますよね。でも、昨日は遅くまで飲んでしまったので、今日は少し眠いです。) |
| GPT-4 | Oh, that's great! **Being energetic in the morning** has a very positive effect on the start of your day, doesn't it? (えー、それはすごいですね!朝から元気なのは、一日の始まりにとても良い影響がありますよね。) |
| TC-GPT-4 | Oh, that's great! **Being energetic in the morning** has a very positive effect on the start of your day, doesn't it? (えー、それはすごいですね!朝から元気なのは、一日の始まりにとても良い影響がありますよね。) |
| Utterance | I still love a big chorus!!!! (やっぱり大合唱は最高だ!!!!) @ 7 p.m (nighttime) |
| TJD | Thanks a lot! (おつかれさまでした!) |
| TC-TJD | Yay! **Thanks for today!** It was fun! (いぇーい!今日はありがとう!楽しかった!) |
| TJD-t | Thanks a lot! I had the best time!!!! (おつかれさまでした!最高にたのしかったです!!!) |
| TC-TJD-t | It was great...! **Thank you so much for today!** (最高だったね...!今日は本当にありがとう!) |
| GPT-3.5 | That's right! That song is a real crowd-pleaser when everyone sings it together. But what about the next song? (そうだね!特にあの曲はみんなで歌うと本当に盛り上がるよね。でも、次の曲はどうする?) |
| TC-GPT-3.5 | Yes, a big chorus is really exciting, isn't it? Are you **going out for a drink afterward**? (そうだね、大合唱は本当に盛り上がるよね。この後は何か飲みに行く予定はある?) |
| GPT-4 | I really agree! It's even more exciting to hear a big chorus **at this time of night**, isn't it? (本当にそうだね!夜のこの時間に大合唱を聴くと、さらに感動が増すよね。) |
| TC-GPT-4 | I really do! But **it's almost time for dinner**, so why don't we head home? (本当にそうだね!でもそろそろ夕食の時間も近いから、帰ろうか?) |

Table 8: Examples of generated responses by each model. TC-$\mathcal{M}$ is time-considerable $\mathcal{M}$ achieved by our proposed reranking method. White and gray rows correspond to $\mathcal{M}$ and TC-$\mathcal{M}$. Time-specific phrases are indicated by **bold**. All utterances and responses were originally written in Japanese and translated into English by the authors.

Table 8 shows example responses $\bar{r}_\mathcal{M}$ and $r^*_\mathcal{M}$ generated by each $\mathcal{M}$ and TC-$\mathcal{M}$, where $\bar{r}_\mathcal{M}$ and $r^*_\mathcal{M}$ are shown in white and gray rows. The result indicates $r^*_\mathcal{M}$ contained more time-specific expressions than $\bar{r}_\mathcal{M}$; e.g., "I overslept" in the morning and "thanks for today" at night. The detailed analysis is described in Appendix A.3. For the first utterance, the original GPT-4 already generated time-specific responses, and TC-GPT-4 selected the same response as its output; namely, in this example, TC-GPT-4 considered that the original response was sufficiently time-considerable.

## 6 Conclusion

We proposed a new reranking method to construct time-considerable dialogue models that distinguish between always natural responses **AN** and temporally natural responses **TN** and actively output **TN**. We verified the assumption *users prefer* **TN** *to* **AN** by human evaluations and introduced a new metric to classify the NUTA categories of responses. We proposed a new reranking method to make existing dialogue models *time-considerable* using the metric and empirically showed that our

method improved the qualities of existing models.

A promising future study is to control the degree to which a time-considerable dialogue model actively uses time information in different situations. Not only time information but all external information is valuable when used appropriately, but excessive use may harm the users' impression. Therefore, we plan to develop a mechanism to estimate an appropriate TD@$t$ value that a response should have in the current situation and to output or generate a response with the estimated value.

## Acknowledgements

This work was supported by JSPS KAKENHI Grant Number 19H05693.

## Limitations

**Effect of TAN's quality** In our proposed reranking method, we used TJD-t, which is a time-aware transformer-based Japanese dialogue model obtained by fine-tuning, as TAN, but our experimental result showed the quality of TJD-t was lower than the other models. One of our contributions is that we empirically showed that our reranking method

successfully improved the S (specificity to time) score of each model even though the quality of TAN (TJD-t) is somewhat low. However, it would be desirable to investigate how the effectiveness of our proposed method changes as the quality of TAN changes.

**Different types of time information**    In this paper, we used the time of day as time information. However, there are other types of time information that have longer periods, such as day of the week and seasons. It is promising future research to investigate how the quality of dialogues changes with the use of such longer periodic time information.

**Cultural differences in time information**    In this paper, we investigate the effect of the use of time information on dialogues in Japanese; however, it has been shown that the time-specific expression varies depending on the country and culture (Shwartz, 2022). Therefore, it is desirable to investigate whether the proposed method can produce time-considerable responses for different languages.

## Ethics Statement

In this paper, we employed workers using a crowdsourcing service. We made sure that the workers were paid above the minimum wage. It applies to all crowdsourcing experiments in this paper.

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

## A  Appendix

### A.1  Dataset Construction

We describe the filtering rules used to obtain the NUTA dataset in Section 3 and the fine-tuning dataset in Section 4.

**NUTA dataset**  In Section 3, we constructed the NUTA dataset which is a collection of tuples of utterances, their spoken times, their responses, and their NUTA categories. As utterances, we extracted Japanese tweets posted between May and December 2022 that satisfied the following conditions:

- Do not contain URLs, usernames, other tweets, parentheses,
- Consists of 6 to 30 characters,
- Not posted by users whose names contain "bot",
- Not Replied to another tweet.

We randomly selected 1,000 tweets from the extracted tweets and manually deleted tweets containing discriminatory, violent, or inappropriate expressions. As a result, we obtained 640 appropriate tweets and used them as utterances $\mathcal{U}$.

**Fine-tuning dataset**  In Section 4, we constructed a fine-tuning dataset in the same manner as Shwartz (2022) to obtain time-aware TJD by training TJD on the obtained dataset. We collected Japanese tweet-reply pairs with their timestamp posted between August 2021 to April 2022 that do not contains URLs or other tweets. As a result, we obtained 470,255,625 triplets.

### A.2  Implementational and Experimental Settings

We here describe the detailed settings of our implementation and experiments.

**TJD and TJD-t**  For a time-unaware dialogue model, we used TJD which is a transformer-based Japanese dialogue model with 1.6B parameters trained on over two billion tweet-reply pairs (Sugiyama et al., 2023). We downloaded the trained TJD [3] and obtained a time-aware TJD, denoted by TJD-t, by fine-tuning TJD on Fairseq [4] (Ott et al., 2019), which is a sequence modeling toolkit to train custom models for various takes including translation, summarization,

---

[3] https://github.com/nttcslab/japanese-dialog-transformers

[4] https://github.com/facebookresearch/fairseq

---

language modeling, and other text generation tasks. In fine-tuning, we used SentencePiece [5] (Kudo and Richardson, 2018) to tokenize utterances and responses written in Japanese. Table 9 shows the hyperparameters we set to in fine-tuning. We used the computational resource of AI Bridging Cloud Infrastructure (ABCI) provided by the National Institute of Advanced Industrial Science and Technology (AIST).

| Configurations | Values |
|---|---|
| Model Architecture | Transformer |
| Pretrained Model | TJD (Sugiyama et al., 2023) |
| Devices | Nvidia V100 GPU |
| Max tokens | 4,000 |
| Optimizer | Adafactor |
| Learning rate | 1e-04 |
| Learning rate scheduler | inverse sqrt |
| Warmup | 10000 |
| weight decay | 0.0 |
| Loss Function | label smoothed cross entropy |

Table 9: Hyper-parameters for fine-tuning

**GPT-3.5/4**  We used GPT-3.5 and GPT-4 (OpenAI, 2023) as state-of-the-art dialogue models. Since architectures and parameters of GPT-3.5/4 were not publicly available, we used OpenAI APIs `gpt-3.5-turbo` and `gpt-4` to generate responses of GPT-3.5/4. We created the following prompt based on a sample prompt for dialogue tasks provided by OpenAI and gave the prompt as input to GPT-3.5/4 to generate a time-aware response.

```
The current time is [hour], and A and B
are having a conversation. Taking into
account the current time, generate the
following B's response to A's utterance.
However, avoid expressions like "it's
[hour] o'clock now."
A: [utterance]
B:
```

**TD@$t$ in our proposed reranking method**  As shown in Eq. (5), TD@$t$ is the maximum of $\text{CN}_{t^{\text{U}}:t}(r \mid u)$ for all possible $t^{\text{U}} \in \mathcal{T}$, and $t^{\text{U}}$ could be very close to spoken time $t$. In such cases, the time dependency of $r$ might be difficult for humans to understand since its naturalness varies in too *short-term*. To avoid detecting such short-term time dependency, we divided 24 hours into three intervals, morning (3 to 8), noon (9 to 17),

---

[5] https://github.com/google/sentencepiece

and night (18 to 2). We used 6, 13, and 22 as the representatives of morning, noon, and night, respectively. The division and their representatives were determined with reference to Yamamoto and Shimada (2019). We computed TD@$t$ by Eq. 5 using the above $\mathcal{T}$ excluding the representative of spoken time $t$. For instance, for $t = 10$, we use $\mathcal{T} = \{6, 22\}$ to evaluate TD@$t$ because the representative of 10 is defined as 13. We conducted the same experiments as Section 4 using the above TD@$t$, and Table 10 and 11 show the results. The results indicate that the accuracy of **TN** classification was slightly improved by introducing the above time intervals but not for the average accuracy. Therefore, it cannot be said that either TD with intervals is better than the original TD.

| | Naturalness | | TD | | PCMI | |
|---|---|---|---|---|---|---|
| | TUN | TAN | @all | @$t$ | @all | @$t$ |
| **N** | 0.60 | 0.68 | 0.48 | 0.61 | 0.48 | 0.55 |
| **T** | 0.54 | 0.54 | 0.65 | 0.58 | 0.51 | 0.48 |

Table 10: Accuracy of **N/U** and **T/N** classifications with the morning/day/naight quantization

| | TD | | PCMI | |
|---|---|---|---|---|
| | @all | @$t$ | @all | @$t$ |
| **AN** | 0.50 | 0.39 | 0.41 | 0.38 |
| **TN** | 0.48 | 0.61 | 0.39 | 0.51 |
| **TU** | 0.41 | 0.15 | 0.30 | 0.14 |
| **AU** | 0.46 | 0.45 | 0.34 | 0.32 |
| Ave. | 0.46 | 0.40 | 0.36 | 0.34 |

Table 11: Accuracy of NUTA classification with the morning/day/naight quantization

### A.3 Time-specific Expressions

We conducted a subjective evaluation to count the number of time-specific expressions by authors. As a result, our method increased their occurrences by 80% and 8% for TJD-t and GPT-4, respectively. These results on occurrences of time-specific expression are consistent with the improvements in S shown in Table 7.