# OpenReview forum: "Time-Considerable Dialogue Models via Reranking by Time Dependency"
_EMNLP/2023/Conference — EMNLP 2023 Findings_

### Official Review · Reviewer_GNUY · 2023-08-04

**Typos Grammar Style And Presentation Improvements:** Page 4, 2nd col. header
**Soundness:** 3

**Excitement:**

3: Ambivalent: It has merits (e.g., it reports state-of-the-art results, the idea is nice), but there are key weaknesses (e.g., it describes incremental work), and it can significantly benefit from another round of revision. However, I won't object to accepting it if my co-reviewers champion it.

**Paper Topic And Main Contributions:**

The authors propose a method to rerank an LLM's responses based on the time of day. They propose a method to classify responses based whether they are always (un)suitable independent of the time of day or temporarily (un)suitable.

**Reasons To Accept:**

Interesting proposal to consider additional context when ranking LLMs' responses.

**Reasons To Reject:**

Too narrow. It seems to be that their proposal can be extended to other contextual information (beyond time of day) and that their algorithms should be extended to consider other context. They should evaluate how their approach varies in accuracy when considering different contexts.

**Reproducibility:**

2: Would be hard pressed to reproduce the results. The contribution depends on data that are simply not available outside the author's institution or consortium; not enough details are provided.

**Reviewer Confidence:**

2: Willing to defend my evaluation, but it is fairly likely that I missed some details, didn't understand some central points, or can't be sure about the novelty of the work.

---

> ### Author Rebuttal · Authors · 2023-08-29
>
> Thank you for taking the time to review our paper and for providing valuable feedback and comments.
> We appreciate the insights and suggestions of the reviewer.
>
> # Reasons To Reject
> > Too narrow. It seems to be that their proposal can be extended to other contextual information (beyond time of day) and that their algorithms should be extended to consider other context. They should evaluate how their approach varies in accuracy when considering different contexts.
>
> We consider that our method is NOT narrow.
> One of our contributions is that this paper distinctly separated contextual information into "persistent information" and "optional information."
> Whereas optional information, such as user information, sometimes exists but is sometimes absent, persistent information, such as time information, always exists.
> This paper primarily focused on time information, an essential example of such persistent information.
> We proposed a new metric for evaluating dependency between utterances and persistent information, while existing metrics are for optional information.
> Moreover, we introduced the NUTA category for responses based on time dependency, which helps analyze utterances with time information.
> Using our new metric and the NUTA category, we proposed a method to generate appropriate responses according to time information without modifying the base model.
> We validated the efficacy of our method through experiments.
> Due to space constraints in the paper, we used the time-of-day interval in our experiments; however, we can apply our method to other time intervals and persistent information.
> As acknowledged by other reviewers, our proposed method must be helpful for future research.
>
>
> # Typos Grammar Style And Presentation Improvements
>
> Thank you for your comment.
> It is a valuable suggestion for improving the readability and organization of the paper, and we will incorporate it in the revision.

---

### Official Review · Reviewer_eL5Z · 2023-08-05

**Soundness:** 3

**Excitement:**

3: Ambivalent: It has merits (e.g., it reports state-of-the-art results, the idea is nice), but there are key weaknesses (e.g., it describes incremental work), and it can significantly benefit from another round of revision. However, I won't object to accepting it if my co-reviewers champion it.

**Paper Topic And Main Contributions:**

This paper investigates how time information can be incorporated into dialogue systems. It introduces corresponding notation and metrics, and shows how to re-rank outputs from 4 dialogue systems for a Japanese dataset to improve time-appropriate responses.

**Questions For The Authors:**


l. 259: how were the "experts" selected, what was their expertise?
l. 512: which prompts were used for GPT models? Can you add them?
l. 524: can you assess the impacts that this usage of time intervals (instead of single hours) had?
l. 541: why did you measure lengths and n-grams? What does diversity have to do with the goal of this paper?
l. 574ff: this holds only for the weaker models, but NOT for GPT-4, wich is out of the box already excellent (SSI-t 0.83). Do you agree?
l. 593: did you count the number of time-specific expressions (if so: where is the data?), or how did you come to this conclusion?

why do you not publish your dataset?


**Reasons To Accept:**

- interesting research question
- introduced metrics are reasonable and helpful for assessing time appropriateness


**Reasons To Reject:**

- the results show that only weak models benefit from the introduced methods, but not the strongest model, GPT-4


**Reproducibility:**

3: Could reproduce the results with some difficulty. The settings of parameters are underspecified or subjectively determined; the training/evaluation data are not widely available.

**Reviewer Confidence:**

3: Pretty sure, but there's a chance I missed something. Although I have a good feel for this area in general, I did not carefully check the paper's details, e.g., the math, experimental design, or novelty.

**Typos Grammar Style And Presentation Improvements:**

l. 101: add remark that you will verify this assumption lateron in the paper
l. 103: an example what time-aware vs time-considerable is would be helpful here
l. 149: evaluated-> evaluate
l. 172-181: could be skipped, was already said before
l. 249: definition what (S) means is unclear
l. 259: use other character (not S) for scores, since S is already used for Specifity of Time, and italic font is hard to distinguish
l. 275: insert "more" before TN
l. 325ff: sentence seems corrupted
l. 355: remove "let"
l. 457: how is delta chosen?
l. 565: there is something missing before M, probably "all"
table 8: grey/white lines are not visible (at least in my black-and-white printout)

overall, the paper is sometimes hard to follow since a huge amount of abbreviations is introduced. Maybe check if some abbreviations are really necessary

---

> ### Author Rebuttal · Authors · 2023-08-29
>
> Thank you for taking the time to review our paper and for providing valuable feedback and comments. We appreciate the insights and suggestions of the reviewer.
>
> # Questions
>
> Question 1:
> > l. 259: how were the "experts" selected, what was their expertise?
>
> The "experts" are in-house workers specialized in annotating and evaluating dialogues and have worked in their positions for at least five years.
>
>
> Question 2:
> > l. 512: which prompts were used for GPT models? Can you add them?
>
> The template prompts used for GPT models are shown in Appendix A.2.
> The detailed prompts were created by replacing the [utterance] and [hour] placeholders in the template with concrete values.
> We omitted the detailed prompts since the number of detailed prompts is the same as the dataset size.
>
>
> Question 3:
> > l. 524: can you assess the impacts that this usage of time intervals (instead of single hours) had?
>
> There are various ways to determine time intervals: hours, morning/noon/night, seasons, and more.
> Whereas a single hour is just a unit of time, morning/noon/night and seasons are defined based on the human lifecycle.
> Thus, the latter is more intuitive for human understanding and used as meaningful intervals in other studies: e.g., Yamamoto et al. used morning/noon/night, and Sato et al. used seasons.
> We used morning/noon/night as a time interval, similar to Yamamoto et al., to provide a more intuitive analysis.
>
>
> Question 4:
> > l. 541: why did you measure lengths and n-grams? What does diversity have to do with the goal of this paper?
>
> Length and distinct (n-grams) are metrics commonly used to measure the diversity of dialogue models.
> Low-diversity models tend to generate monotonous responses and are often considered low-quality.
> Our paper aims to generate responses with high time dependency while avoiding the side effects of sacrificing diversity.
> Thus, we calculated lengths and distinct (n-grams) to ensure our method maintained the diversity of the base models, and the experimental results showed that our method did not cause a reduction in diversity.
>
>
> Question 5:
> > l. 574ff: This holds only for the weaker models, but NOT for GPT-4, which is out of the box already excellent (SSI-t 0.83). Do you agree?
>
> As the reviewer pointed out, applying our proposed method may not enhance the naturalness of stronger models such as GPT-4.
> This paper aims to increase responses' "specificity to time" while maintaining their naturalness, and our method successfully increased S (specificity to time) of all models, including GPT-4, without decreasing their naturalness.
> The result implies that our method is effective not only for weaker models but also for stronger models like GPT-4.
>
>
> Question 6:
> > l. 593: did you count the number of time-specific expressions (if so: where is the data?), or how did you come to this conclusion?
>
> As shown in Table 7, our method improved S scores by 66.7% and 4.5% for TJD-t and GPT-4, respectively.
> We conducted a subjective evaluation to count the number of time-specific expressions by authors.
> As a result, our method increased their occurrences by 80% and 8% for TJD-t and GPT-4, respectively.
> Since these results on occurrences of time-specific expression are consistent with the improvements in S shown in Table 7, we omitted the results in our paper for brevity.
> Given the reviewer's comment, we consider adding the results in Appendix.
>
>
> Question 7:
> > why do you not publish your dataset?
>
> The Twitter API policy does NOT allow us to redistribute the dataset constructed by the Twitter API.
> Thus, we cannot publish the dataset we used in this paper.
>
>
> # Reasons To Reject
> > the results show that only weak models benefit from the introduced methods, but not the strongest model, GPT-4
>
> As mentioned in the answer to Question 5, our method improved S not only for weak models but also for strong models such as GPT-4.
> Consequently, GPT-4 would also benefit from our reranking method to improve the "specificity to time" of its responses.
>
>
> # Typos Grammar Style And Presentation Improvements
>
> Thank you for your constructive comments.
> They are valuable suggestions for improving the readability and organization of the paper, and we will incorporate them in the revision.
>
>
> Presentation Improvement 1:
> > l. 103: an example what time-aware vs time-considerable is would be helpful here
>
> Figure 1 shows four examples of AN, TN, TU, and AU responses.
> The reviewer pointed out that providing another example that compares explicitly AN and TN is beneficial for readers to understand our idea.
> We plan to add such an example to our paper.
>
>
> Presentation Improvement 2:
> > l. 457: how is delta chosen?
>
> Delta is a threshold to filter generated responses based on their naturalness; we can remove responses with lower naturalness by setting a higher value to the delta.
> Therefore, when the response generation model is weak, this filtering mechanism can improve the naturalness of the generated responses.
> We set the delta value to 0 because we used sufficiently strong models for response generation and did not aim to improve their naturalness.

---

### Official Review · Reviewer_Ytjk · 2023-08-05

**Soundness:** 3

**Excitement:**

4: Strong: This paper deepens the understanding of some phenomenon or lowers the barriers to an existing research direction.

**Paper Topic And Main Contributions:**

This paper addressed generating more attractive responses by using external information beyond the past context of the dialogue, specifically, using time information to generate time-aware responses. After experimentally confirming the plausibility of the hypothesis that "people prefer responses that are natural only at specific times to those that are natural always," they proposed a reranking method for preferentially generating "responses that are natural only at specific times." Experiments showed the usefulness of the proposed method when applied to a standard generative model (e.g., GPT-4) by automatic and human evaluation.

**Questions For The Authors:**

- Regarding L.257-263, what was the degree of agreement among the annotators?
- I agree with the advantage of the reranking method (e.g., easy to apply to existing models). On the other hand, how much does the reranking method improve the accuracy of the task compared to existing methods that straightforwardly train to generate time-aware responses, such as the method of Sato et al. mentioned by the authors in L153?

**Reasons To Accept:**

- The challenge to develop the time-considerable dialogue models is interesting.
- The NUTA category, consisting of four types (AN, TN, TU, AU), proposed to discuss the validity of responses in a time-aware manner, which will be useful for future research.
- The careful notation and mathematical explanation make it easy to follow the description of the proposed method and the discussion.

**Reasons To Reject:**

- The experiments were conducted using only Twitter-sourced data. Due to the difficulty of redistributing the tweet data and the fact that the Twitter API is currently unavailable, it would not be easy to reproduce or follow up on this paper. Lack of reproducibility is a serious problem in scientific papers, and I am concerned about this point.

**Reproducibility:**

3: Could reproduce the results with some difficulty. The settings of parameters are underspecified or subjectively determined; the training/evaluation data are not widely available.

**Reviewer Confidence:**

3: Pretty sure, but there's a chance I missed something. Although I have a good feel for this area in general, I did not carefully check the paper's details, e.g., the math, experimental design, or novelty.

**Typos Grammar Style And Presentation Improvements:**

- If I understand correctly, the actual data the authors have created is only in Japanese. If that is the case, including the actual Japanese data as examples (especially figures and tables) in the paper would be accurate. Then, it would be desirable to include an English translation to benefit many readers.
 - L.517: Please define N, \delta, p, respectively.

---

> ### Author Rebuttal · Authors · 2023-08-29
>
> Thank you for taking the time to review our paper and for providing valuable feedback and comments.
> We appreciate the insights and suggestions of the reviewer.
>
> # Questions
>
> Question 1:
> > Regarding L.257-263, what was the degree of agreement among the annotators?
>
> Two annotators evaluated 100 responses for the four scores (SU, ST, S, and I) with either 0 or 1.
> The degree of agreement among the two annotators was as follows: SU at 0.97, ST at 0.86, S at 0.83, and I at 0.71.
> This agreement rates high enough to trust the annotated scores.
> The agreement rate for I (interestingness) is slightly lower than for the other indicators because interestingness judgments are more subjective than others.
>
>
> Question 2:
> > I agree with the advantage of the reranking method (e.g., easy to apply to existing models). On the other hand, how much does the reranking method improve the accuracy of the task compared to existing methods that straightforwardly train to generate time-aware responses, such as the method of Sato et al. mentioned by the authors in L153?
>
> TJD-t is a model trained in the same manner as Sato et al., an existing method that straightforwardly trains to generate time-aware responses. TC-TJD-t is a model obtained by applying our reranking method to TJD-t.
> Table 7 shows that TC-TJD-t improved all scores (ST, SU, S, I, and SSI-t) compared to the original TJD-t.
> These results indicate that our reranking method successfully improved a model that was straightforwardly trained to generate time-aware responses.
>
>
> # Reasons To Reject
> > The experiments were conducted using only Twitter-sourced data. Due to the difficulty of redistributing the tweet data and the fact that the Twitter API is currently unavailable, it would not be easy to reproduce or follow up on this paper. Lack of reproducibility is a serious problem in scientific papers, and I am concerned about this point.
>
> As the reviewer pointed out, the free Twitter API is unavailable, making it difficult to collect tweets without incurring costs.
> However, Twitter Inc. offers a paid Twitter API to collect tweets, including those we used in this paper.
> Consequently, everyone can collect the same dataset by the paid API with our filtering rules described in Appendix A.1.
> Furthermore, all base models used in our experiments are publicly available.
> Given these points, it is NOT difficult to reproduce or follow up on this paper.
> We will consider releasing our code If the reviewer's concern is reproducibility; however, we are not allowed to release the Twitter dataset we constructed in this paper due to its terms of use.
>
>
> # Typos Grammar Style And Presentation Improvements
>
> Thank you for your constructive comments.
> They are valuable suggestions for improving the readability and organization of the paper, and we will incorporate them in the revision.

---

### Meta-Review · Area_Chair_NEqX · 2023-09-20

**Recommendation:** 3

**Metareview:**

The authors propose a method to add time awareness to conversational models. The reviewers generally like the approach and the importance of the problem. However, the reviewers also point out that improvements using this method are only found in smaller models, that the data used is proprietary, and that the contribution may be too narrow. The authors respond to most of the reviewers concerns and I think this would be a good Findings paper.

---

### Decision · Program_Chairs · 2023-10-07

**Decision:**

Accept-Findings

**Comment:**

The authors propose a method to add time awareness to conversational models. The reviewers generally like the approach and the importance of the problem. However, the reviewers also point out that improvements using this method are only found in smaller models, that the data used is proprietary, and that the contribution may be too narrow. The authors respond to most of the reviewers concerns and I think this would be a good Findings paper.